Masticatory biomechanics of the Laotian rock rat, Laonastes aenigmamus, and the function of the zygomaticomandibularis muscle

Cox Philip G. 1 philip.cox@hyms.ac.uk
Kirkham Joanna 2 3
Herrel Anthony 4
1 Centre for Anatomical and Human Sciences, Hull York Medical School , University of Hull, Hull , UK
2 Centre for Anatomical and Human Sciences, Hull York Medical School, University of York, York , UK
3 College of Medical and Dental Sciences, University of Birmingham , Birmingham , UK
4 Ecologie et Gestion de la Biodiversite, CNRS/MNHN , Paris , France
Jungers William
Electronic publication date: 2013 Sep 12
Publication date: 2013
Volume: 1
Electronic Location ID: e160
Received 2013 Aug 1; Accepted 2013 Aug 23
Copyright: © 2013 Cox et al.
Copyright year: 2013
Copyright holder: Cox et al.
License: This is an open access article distributed under the terms of the Creative Commons Attribution License, which permits unrestricted use, distribution, and reproduction in any medium, provided the original author and source are credited.
License URL: https://creativecommons.org/licenses/by/3.0/

Keywords: Masticatory muscles, Feeding, Rodentia, Skull, Biomechanics, Hystricomorphy, Finite element analysis

Funding: UK National Health Service The funding for Joanna Kirkham’s intercalated BSc was provided by the UK National Health Service. The funders had no role in study design, data collection and analysis, decision to publish, or preparation of the manuscript.

==============================
The Laotian rock rat, Laonastes aenigmamus, is one of the most recently discovered species of rodent, and displays a cranial morphology that is highly specialised. The rostrum of L. aenigmamus is exceptionally elongate and bears a large attachment site for the infraorbital portion of the zygomaticomandibularis muscle (IOZM), which is particularly well-developed in this species. In this study, we used finite element analysis to investigate the biomechanical performance of the Laotian rock rat cranium and to elucidate the function of the IOZM. A finite element model of the skull of L. aenigmamus was constructed and solved for biting on each of the teeth (incisors, premolar and molars). Further load cases were created and solved in which the origin of the IOZM had been moved anteriorly and posteriorly along the rostrum. Finally, a set of load cases were produced in which the IOZM was removed entirely, and its force was redistributed between the remaining masticatory muscles. The analysis showed that, during biting, the most stressed areas of the skull were the zygomatic and orbital regions. Compared to other rodents, L. aenigmamus is highly efficient at incisor gnawing, but less efficient at molar chewing. However, a relatively constant bite force across the molar tooth row may be an adaptation to folivory. Movement of the origin of the IOZM had little on the patterns of von Mises stresses, or the overall stress experienced by the cranium. However, removal of the IOZM had a substantial effect on the total deformation experienced by the skull. In addition, the positioning and presence of the IOZM had large impact on bite force. Moving the IOZM origin to the anterior tip of the rostrum led to a substantially reduced bite force at all teeth. This was hypothesised to be a result of the increasing horizontal component to the pull of this muscle as it is moved anteriorly along the rostrum. Removal of the IOZM also resulted in reduced bite force, even when the total input muscle force was maintained at the same level. It was thus concluded that the function of the IOZM in L. aenigmamus is to increase bite force whilst reducing cranial deformation. If the IOZM can be shown to have this function in other rodent groups, this may help explain the evolution of this muscle, and may also provide an understanding of why it has evolved independently several times within rodents.

Introduction

The Rodentia is the most speciose of all mammalian orders, with over 2,200 extant species (Wilson & Reeder, 2005) in addition to a large number of fossil forms. Despite such taxonomic diversity, morphological variation within the order is relatively limited, particularly with regard to the skull and mandible (Wood, 1965; Hautier et al., 2011). Hence, determining relationships between rodent species based on morphology is difficult, and rodent taxonomy has been historically controversial, with two competing classifications arising in the second half of the 19th century. First, Brandt (1855) split the rodents into three suborders (Sciuromorpha, Hystricomorpha and Myomorpha) based on the morphology of the masseter muscle and its attachment to the rostrum (see Cox & Jeffery, 2011 for details). Later, Tullberg (1899) divided the rodents into two groups (Sciurognathi and Hystricognathi) based on the position of the angular process of the mandible relative to the incisor. Recent molecular phylogenies (Blanga-Kanfi et al., 2009; Churakov et al., 2010; Fabre et al., 2012) have shown that neither of these classifications accurately resolves the evolutionary relationships between the rodents. The three suborders of Brandt (1855), also used by Simpson (1945), have now been discarded, as they are thought to represent polyphyletic groupings of rodents, although the names have been retained in their adjectival form (sciuromorphous, etc.) to describe the three morphotypes of the skull and masseter (Wood, 1965; Cox & Jeffery, 2011). The classification of Tullberg (1899) has fared slightly better, with the Hystricognathi still recognised as a monophyletic clade (the Sciurognathi is paraphyletic with respect to the Hystricognathi). However, recent work (Hautier et al., 2011) has questioned the usefulness of sciurognathy and hystricognathy as morphological terms, noting that there exists a continuum of mandibular forms rather than two discrete morphologies, and that some members of the Hystricognathi actually have mandibles that appear to be almost sciurognathous in form.

The Laotian rock rat, Laonastes aenigmamus (Jenkins et al., 2005), is a recently discovered species of rodent from Southeast Asia. It shows an unusual mixture of cranial, mandibular and muscular morphologies, combining a large part of the zygomaticomandibularis muscle that extends through the enlarged infraorbital foramen to attach to the rostrum (Hautier & Saksiri, 2009) with a weak lateral displacement of the angular process of the mandible (Hautier et al., 2011). Thus, the Laotian rock rat brings together a hystricomorphous skull and masseter with a lower jaw that is intermediate between sciurognathous and hystricognathous. This combination of characters has made its phylogenetic relationships difficult to ascertain. When first described, a new family, the Laonastidae, was created to house L. aenigmamus (Jenkins et al., 2005). This family was placed within the Hystricognathi as the sister-group to African mole-rats (Bathyergidae) or the dassie rat (Petromuridae). A subsequent analysis (Dawson et al., 2006) showed that L. aenigmamus was in fact a member of the Diatomyidae, a family of rodents previously thought to have gone extinct in the Miocene. Further work then showed the Laotian rock rat to be the sister-taxon to the Ctenodactylidae (Huchon et al., 2007), a family of rodents that also display the combination of a hystricomorphous skull with a sciurognathous mandible (Hautier, 2010). The Ctenodactylidae and Diatomyidae together form the sister-group to Hystricognathi, within the more inclusive clade Ctenohystrica (Fabre et al., 2012).

One particularly notable characteristic of the Laotian rock rat is the morphology of the zygomaticomandibularis muscle. This muscle is the innermost layer of the masseter, and in the hystricomorph condition extends anteriorly through the orbit and the grossly enlarged infraorbital foramen to take an attachment on the rostrum (Wood, 1965; Cox & Jeffery, 2011). In L. aenigmamus, the infraorbital portion of the zygomaticomandibularis (IOZM) is especially well-developed, extending over halfway along the rostrum (Hautier & Saksiri, 2009), which is itself exceptionally elongated compared to other rodents (Herrel et al., 2012). Various functions have been proposed for the IOZM. Becht (1953) noted that its origin on the rostrum between the incisors and cheek teeth would enable it to function as a second class lever during chewing at the molars, but as a third class lever during gnawing at the incisors. Thus, it can produce both fine control at the incisors and strong pressure at the molars. Using electromyography, Weijs & Dantuma (1975) found that the IOZM was firing at low intensities during jaw opening in rats, and suggested that it may have a role in the fine control of the opening phase of mastication. With regard to L. aenigmamus specifically, Herrel et al. (2012) proposed that the strong development of the IOZM and its anterior origin on the rostrum would produce a strong horizontal force component to the bite. This would result in optimal functioning at low gape angles and the generation of uniform force along the tooth row, both of which would be beneficial for a folivorous diet, which has been suggested for the Laotian rock rat (Scopin et al., 2011).

The aim of this study is to investigate the biomechanics of feeding in Laonastes aenigmamus. Finite element analysis (FEA) will be used to examine the response of the cranium during gnawing at the incisors and chewing at the molars. FEA is a computer-based engineering technique that enables the prediction of stress, strain and deformation in a complex geometric object subjected to a load (Rayfield, 2007). It has been successfully used to study the biomechanics of feeding in a number of mammalian groups (e.g. Dumont, Piccirillo & Grosse, 2005; Kupczik et al., 2007; Wroe, 2010; Bright & Rayfield, 2011) including rodents (Cox et al., 2011; Cox et al., 2012). The major advantage of FEA in the study of biological systems is that elements such as muscle or bone can be modified or removed at will, without the practical and ethical concerns that would arise with in vivo work. In this study, the contribution of the IOZM to feeding will be investigated by changing its attachment site and removing it altogether. In this way, we aim to understand the function of the IOZM, which is so highly developed in the Laotian rock rat, and to elucidate why such an unusual morphology has evolved in this species.

Materials & Methods

Sample and model creation

MicroCT scans of an adult female Laonastes aenigmamus (specimen number KY213), previously obtained for an earlier research project (Herrel et al., 2012), were provided by AH. Voxels were isometric and the voxel resolution was 0.137 mm. Further details of the scanning protocol can be found in Herrel et al. (2012).

Avizo 7.0 (Visualization Sciences Group, Burlington, MA, USA) was used to create a 3D volume reconstruction of the cranium of the Laotian rock rat. The bone and teeth were segmented separately so that different elastic properties could be assigned to them. Within the incisors, the enamel, dentine and pulp were not differentiated, as the scan resolution was not sufficient to distinguish these materials from one another. In addition, varying the material properties of these components has been shown to make little difference to the overall deformation of the skull (Cox & Jeffery, 2011). For similar reasons, the periodontal ligament was not included (Wood et al., 2011). The model was converted to an eight-noded FE mesh by direct voxel conversion, using VOX-FE, custom-built FE software (Liu et al., 2012), resulting in a model of 1734787 elements.

Material properties, constraints and loads

Bone was assigned a Young’s modulus of 17 GPa, and the teeth were given a Young’s modulus of 30 GPa. Both materials were modelled to be linearly elastic and isotropic with a Poisson’s ratio of 0.3. These values were based on previous nano-indentation work on rodents (Cox et al., 2012) and FE studies on other mammals (Kupczik et al., 2007).

The model was constrained at three locations: the left and right temporo-mandibular joints (TMJ) and the biting tooth. The TMJs were constrained on the ventral surface of the zygomatic process of the squamosal in all three dimensions. However, the biting tooth was only constrained in the dorsoventral axis (i.e., perpendicular to the occlusal plane). The number of nodes constrained at each location varied between 191 and 221.

Loads were added to the model to represent the forces generated by the following muscles: superficial masseter; deep masseter; zygomaticomandibularis (anterior, posterior and infraorbital parts); temporalis (main, orbital and posterior parts); internal pterygoid; and external pterygoid. Muscle attachment sites and directions of pull were assigned based on the detailed dissections presented in Hautier & Saksiri (2009) and are shown in Fig. 1. Table 1 gives the muscle mass and mean fibre length of each masticatory muscle, measured from the dissection of specimen KY213. Muscle masses were converted to volumes, assuming a muscle density of 1.0564 g cm−3 (Murphy & Beardsley, 1974), and the physiological cross-sectional area (PCSA) of each muscle (given in Table 1) was calculated by dividing the muscle volume by mean fibre length. Muscle forces (Table 2) were calculated by multiplying PCSAs by an intrinsic muscle stress value of 0.3 N mm−2 (van Spronsen et al., 1989). In order to investigate the function of the IOZM, alternate versions of the models were created with the origin of this muscle moved anteriorly and posteriorly, and also with the IOZM omitted completely on each side. In the models with a posteriorly shifted IOZM, the insertion point of the IOZM was kept the same, resulting in a more vertically directed muscle vector. However, in the models with an anteriorly shifted IOZM, the insertion point was moved to reflect the wrapping of the muscle around the zygomatic process of the maxilla, resulting in a highly horizontally directed muscle vector (see Fig. 1). To facilitate comparisons between models with and without the IOZM whilst retaining the same overall input muscle load, where it was omitted, the force of the IOZM was redistributed across the remaining masticatory muscles. This was done in such a way as to preserve their relative proportions. The muscle forces applied in the absence of the IOZM are given in Table 2.

Figure 1 Attachment sites and orientations of muscle loads applied to FE model.

Skull of Laonastes aenigmamus shown in (A) lateral and (B) ventral view. AZM, anterior zygomaticomandibularis; DM, deep masseter; EP, external pterygoid; IOZM, infraorbital part of the zygomaticomandibularis; IOZMa, anterior placement of IOZM origin; IOZMp, posterior placement of IOZM origin; IP, internal pterygoid; MT, main part of termporalis; OT, orbital part of temporalis; PT, posterior part of temporalis; SM, superficial masseter. Posterior zygomaticomandibularis not shown.

Table 1 Muscle mass (g), mean fibre length (mm) and PCSA (cm2) of masticatory muscles of Laonastes aenigmamus, specimen KY213.

	Mass (g)	Fibre length (mm)	PCSA (cm2)	
Superficial masseter	0.348	8.73	0.376	
Deep masseter	0.235	7.71	0.287	
Anterior ZM	0.285	9.16	0.293	
Posterior ZM	0.033	3.70	0.084	
Infraorbital ZM	0.152	9.71	0.148	
Main temporalis	0.047	8.58	0.052	
Orbital temporalis	0.019	4.67	0.037	
Posterior temporalis	0.017	3.70	0.042	
Internal pterygoid	0.188	6.34	0.280	
External pterygoid	0.118	4.67	0.239	
Notes.

ZM zygomaticomandibularis

Table 2 Muscle loads (N) applied to each side of each model in the presence and absence of the IOZM.

	With IOZM	Without IOZM	
Superficial masseter	11.31	12.30	
Deep masseter	8.62	9.37	
Anterior ZM	8.80	9.57	
Posterior ZM	2.50	2.72	
Infraorbital ZM	4.44	—	
Main temporalis	1.56	1.70	
Orbital temporalis	1.12	1.21	
Posterior temporalis	1.27	1.38	
Internal pterygoid	8.40	9.13	
External pterygoid	7.15	7.78	
Total	55.17	55.17	
Notes.

ZM zygomaticomandibularis

Model solution and analysis

The finite element model of Laonastes aenigmamus was solved for biting at each tooth along the dental arcade, using VOX-FE. Gnawing was assumed always to be bilateral, as a result of the close apposition of the incisors, whereas chewing at the premolars and molars was modelled unilaterally on the left side. Von Mises stress patterns across the skull and bite forces at the teeth were recorded from the solved models. Following Cox & Jeffery (2011) and O’Higgins et al. (2011), geometric morphometrics was used to study the deformation patterns across the cranium. Thus, a set of landmarks (3D co-ordinate data) was recorded from each loaded skull, as well as from the original unsolved model. The landmark set was partially based on that used in Cox & Jeffery (2011) and is shown in Fig. 2. Landmarks were concentrated in the rostral, orbital and zygomatic regions, as these were the areas experiencing the highest strains. The landmarks were then subjected to a geometric morphometric analysis using MorphoJ software (Klingenberg, 2011). This consisted of co-registering the landmarks via Procrustes superimposition followed by a principal components analysis (PCA). Cranial deformations along PC axes were visualised using the EVAN toolbox (http://www.evan-society.org).

Figure 2 Landmarks used in GMM analysis of skull deformations.

Reconstruction of skull of Laonastes aenigmamus in (A) dorsal, (B) ventral and (C) left lateral view. 1, anteriormost point on internasal suture; 2, midpoint on cranium between anterior roots of zygomatic arch; 3, midpoint between medialmost points on orbital margins; 4, midpoint on skull roof between zygomatic processes of the squamosal; 5, posteriormost point in dorsal midline; 6, midpoint between ventral margins of incisal alveoli; 7, midpoint between anteriormost points of premolars; 8, posteriormost midline point on palate; 9, midpoint between posterior margins of pterygoid flanges; 10, ventralmost point on margin of foramen magnum; 11, anteriormost point on naso-frontal suture; 12, dorsalmost point on incisal alveolar margin; 13, rostralmost point of infraorbital fossa; 14, midpoint between incisor and premolar on ventrolateral rostral margin; 15, midpoint of dorsal margin of infraorbital fossa; 16, midpoint between 15 and 17; 17, anteriormost attachment of zygomatic arch to rostrum; 18; posteriormost point of infraorbital margin; 19, point on ventrolateral margin of zygomatic arch in same coronal plane as midpoint of M1; 20, apex of tubercle on anterior orbital margin; 21, dorsalmost point on orbital margin; 22, ventralmost point on orbital margin; 23, midpoint between 21 and 24; 24, posteriormost point on orbital margin. Landmarks 11-24 recorded on both sides.

The results generated by the Laotian rock rat model were compared to those found in other rodents, such as squirrels, guinea pigs and rats (Cox et al., 2012). As it’s difficult to compare absolute bite force between models of different sizes, the mechanical efficiency of biting was calculated. Mechanical efficiency of biting is the ratio of output bite force to input muscle force, and represents the proportion of muscle force that is translated into bite force, i.e., is not lost to deformation of the mandible or generation of joint reaction force at the condyles (Dumont et al., 2011). As it is a proportion, mechanical efficiency is size-independent and facilitates clearer comparisons between skulls of varying sizes.

Results

Figure 3 shows the von Mises stress patterns generated across the skull of L. aenigmamus during biting at the incisor, first molar and third molar. It can be seen that, aside from the biting tooth, the zygomatic arch is the most stressed region of the skull, followed by the orbital region. The rostrum experiences a moderate degree of stress during incisor gnawing, but is unstressed during molar chewing, and the occipital region is unstressed during all bites. From visual inspection of the stress distribution figures it is difficult to determine a great deal of variation between bites at different points along the tooth row, even between incisors and molars. However, by studying the mean von Mises stresses across the skull (Table 3) it can be seen that there are indeed subtle differences between bites on different teeth. In general, overall stress increases as the bite point moves closer to the jaw articulation from the premolar to the third molar. However, the incisor bite does not fit this trend, and shows the greatest mean stress across the skull of all bites, presumably because the rostrum is stressed in incisor bites but not in molar bites.

Figure 3 Predicted distribution of von Mises stresses across the skull of Laonastes aenigmamus.

Arrows indicate the biting tooth. First column, incisor bites; second column, M1 bites; third column, M3 bites. First line, original models; second line, origin of IOZM moved anteriorly to front of rostrum; third line, origin of IOZM moved posteriorly to back of rostrum; fourth line, IOZM muscle force removed and redistributed proportionally between the remaining masticatory muscles.

Table 3 Mean von Mises stress (MPa) across the skull.

	Original model	IOZM origin moved anteriorly	IOZM origin moved posteriorly	IOZM force redistributed	
I	2.31	2.27	2.32	2.34	
PM	2.14	2.15	2.13	2.15	
M1	2.29	2.28	2.23	2.28	
M2	2.66	2.59	2.60	2.57	
M3	3.04	2.86	2.99	2.95	
Notes.

I incisor

PM premolar

M molar

The maximum bite force predicted at each tooth is given in Table 4. As would be expected from simple lever mechanics, bite force increases the closer the bite point (the biting tooth) is to the fulcrum (the TMJ). By dividing the bite forces by total input muscle force, the mechanical efficiency of biting has been calculated and can be compared with previous analyses on other rodents (Cox et al., 2012). It can be seen (Fig. 4) that the Laotian rock rat performs well at the incisors and premolars, outcompeting the squirrel and guinea pig, and even matching the rat for efficiency at the incisors. However, compared to the other rodents, the mechanical efficiency tails off towards the distal molars and at the third molar, L. aenigmamus has the least efficient bite.

Table 4 Predicted bite forces (N) across the skull.

	Original model	IOZM origin moved anteriorly	IOZM origin moved posteriorly	IOZM force redistributed	
I	29.27	25.74	29.05	26.61	
PM	53.74	46.93	53.72	48.80	
M1	61.00	54.93	61.70	55.78	
M2	68.97	61.92	70.32	62.05	
M3	73.93	65.35	74.38	66.61	
Notes.

I incisor

PM premolar

M molar

Figure 4 Mechanical efficiency of biting at each tooth.

Predicted from FE models of squirrel, guinea pig, rat and Laotian rock rat skulls. Data for squirrel, guinea pig and rat from Cox et al. (2012). I, incisor; PM, premolar (absent in rats); M, molar.

The impact of the IOZM muscle was investigated by altering the position of its origin on the rostrum in VOX-FE. Figure 3 shows the von Mises stress patterns generated across the skull in models with the origin of the IOZM moved anteriorly and posteriorly, as well as in its original position. Despite quite large changes in muscle origins, very few differences can be seen between the models. This situation is confirmed by examining the mean von Mises stresses across the skull (Table 3). It can be seen that moving the IOZM origin anteriorly or posteriorly results in very little change (less than 6%) in mean von Mises stress during both incisor gnawing and molar biting. Although not presented here, the same lack of difference was found on examination of the principal strains across the skull. Predicted bite force was also affected very little by moving the IOZM origin caudally to the most posterior part of the rostrum. However, Table 4 shows that an anterior shift of the IOZM origin resulted in quite a substantial reduction in bite force (between 10 and 13%).

The effect of removing the IOZM muscle (and redistributing its force between the remaining masticatory muscles) can be seen in Fig. 3. As when the IOZM origin was moved, few differences could be detected between the von Mises stress patterns by visual inspection alone, and again, this assertion is supported by examination of the mean von Mises stresses across the skull (Table 3). However, it can be seen from Table 4 that the action of the IOZM has a noticeable effect on bite force. It is clear that the presence of the IOZM enables L. aenigmamus to generate a greater bite force than if it were absent, even when the total input muscle force is the same. The reduction in bite force resulting from removal of the IOZM is approximately 10% for bites on both incisors and molars, which is similar to the effect of shifting the origin of this muscle to the anterior part of the rostrum.

In order to analyse the subtle differences between the deformation patterns generated by the various load cases described above, a geometric morphometric analysis was performed. Landmark data from the original solved models, the models with the IOZM moved anteriorly and posteriorly, and models with the IOZM removed and its force redistributed, as well as landmarks from the unsolved model, were all subjected to Procrustes superimposition and PCA. The plot of the first two principal components (together comprising over 98% of the variation; PC1, 63.2%; PC2, 35.1%) is shown in Fig. 5. The first principal component largely separates the unloaded model from the solved load cases. The point representing the unloaded model (star symbol) is on the far left of the plot whilst the points representing the loaded models are spread out down the right hand side of the diagram. The warped reconstructions indicate that the difference in deformation between the loaded and unloaded models is largely concentrated in the zygomatic region (also shown in Fig. 3). The incisor bites (squares) are clearly separated from bites on the other teeth on PC2. The cheek teeth bites are more closely grouped together, but separable into bites on each of the different teeth and positioned in order along the second principal component from the premolar (diamonds) to the third molar (lines). It can be seen from the reconstructions at the extremes of PC2 that incisor bites tend to deflect the rostrum dorsally (relative to the orbito-temporal region), but that molar bites lead to a dorsal movement of the orbital region (relative to the rostrum). Within bites on each tooth, the models with the IOZM origin in its original position (blue symbols), moved anteriorly (green symbols) and moved posteriorly (red symbols) all group together closely at a similar distance from the unloaded model. This demonstrates that a similar amount of deformation is occurring in each of these load cases. However, the symbols representing the models with the IOZM muscle force redistributed between the other masticatory muscles (orange symbols), are positioned further from the unloaded model than the other load cases, indicating that even more deformation is occurring in these models. As the displacement of these symbols is along PC1, it can be seen that the redistribution of the IOZM is leading to greater deformation mainly in the zygomatic region.

Figure 5 GMM analysis of cranial deformations in Laonastes aengimamus.

Plot of the first two principal components from a GMM analysis of 24 cranial landmarks. Cranial reconstructions and thin-plate splines indicate shape changes ( × 200) along the first and second principal components. PC1 and PC2 not to same scale. Key: star, unloaded model; squares, incisor bites; diamonds, premolar bites; triangles, M1 bites; circles, M2 bites; lines, M3 bites; blue, IOZM in original position; green, IOZM moved anteriorly; red, IOZM moved posteriorly; orange, IOZM force redistributed between other masticatory muscles.

Discussion

A finite element model of the skull of the Laotian rock rat, Laonastes aenigmamus, was created, loaded, constrained and solved. It was shown that the area of the skull experiencing the highest levels of stress was the zygomatic arch, including the zygomatic processes of the maxillary and frontal bones, which is likely to be a result of the large amount of masticatory musculature that attaches directly to this area. Similarly high zygomatic stresses have been noted in other rodents (Cox et al., 2012) as well as in other mammalian groups (Dumont, Piccirillo & Grosse, 2005; Bright & Rayfield, 2011; Dumont et al., 2011). It has been suggested that, in primates, the downward pull of the masseter muscle on the zygomatic arch may be counterbalanced to some degree by the upward pull of a soft tissue structure, namely the temporal fascia (Curtis et al., 2011). Despite many careful dissections (e.g., Baverstock, Jeffery & Cobb, 2013), no temporal fascia has been found in rodents, and this is also true of L. aenigmamus (Hautier & Saksiri, 2009). Thus, for the time being, it must be assumed that, although the zygomatic stresses are high in L. aegnimamus, they are not so high as to pose a danger of bone fracture.

The bite forces predicted in this study demonstrate that the skull of L. aenigmamus can generate bites of 29 N during gnawing and between 53 and 74 N during chewing. By dividing these values by the total input muscle force, the mechanical efficiency of biting was calculated. It was shown that the Laotian rock rat is particularly efficient at incisor biting, having a mechanical efficiency greater than squirrels or guinea pigs, and similar to that of rats (Cox et al., 2012), but is less efficient at molar bites compared to these three rodent species. This would seem to indicate that L. aenigmamus is well-adapted for gnawing and less so for chewing, a conclusion that at first glance appears somewhat at odds with the suggested diet of this species, which is thought to be largely folivorous (Scopin et al., 2011). However, Onoda et al. (2011) have proposed that the generation of a uniform bite force across the tooth row may be beneficial in the processing of fibrous plant material such as leaves. Viewed in this light, L. aenigmamus is well-adapted to folivory with a difference of just 20 N between its premolar and M3 bites, compared to 30–35 N in squirrels and guinea pigs.

One of the more intriguing results of this study is that the position of the IOZM muscle has little effect on the overall stress experienced by the skull of L. aenigmamus. It can be seen from both Fig. 3 and Table 3 that neither the pattern of von Mises stresses nor the mean stress across the skull are greatly affected by moving the origin of the IOZM forwards or backwards. This may be because the IOZM is contributing a relatively small proportion of the total muscle force (around 8%). Although the IOZM appears to be quite a large muscle in the lateral view (see Hautier & Saksiri, 2009), it is relatively small compared to the superficial and deep masseters. Moreover, it also possesses the longest muscle fibres of all masticatory muscles, and so its PCSA (and hence muscle force) is comparatively reduced (Table 1). Thus, its ability to impact the overall stress patterns across the skull may be fairly limited and, furthermore, these patterns may be more strongly influenced by skull morphology than by muscle geometry.

In contrast to the position of its attachment, the presence or absence of the IOZM has a much stronger effect on the cranial biomechanics of L. aenigmamus. Figure 5 shows that the symbols representing the models without the IOZM are positioned further from the unloaded model that the other load cases, indicating that the removal of the IOZM leads to greater deformation (and therefore strain) across the skull. Thus, one of the major functions of the IOZM, at least in L. aenigmamus, appears to be to minimise strain during feeding. This conclusion holds true at all bites, both gnawing and chewing.

In addition to its effect on cranial deformation, it was also found that the IOZM has a strong impact on the bite force produced by L. aenigmamus (Table 4). Specifically, removing the IOZM altogether (and redistributing its force between the remaining masticatory muscles) reduces the bite force generated at all teeth. Similarly, moving the origin of the IOZM to the anterior tip of the rostrum reduces bite force to a similar degree. It is likely that this effect is a result of the wrapping of the IOZM around the zygomatic process of the maxilla, which means that as the IOZM is moved forward, its vector of pull becomes more horizontal (Fig. 1), and so less able to generate bite force. On the other hand, moving the origin of the IOZM posteriorly appears to have little effect on bite force. Interestingly, the reduction in bite force resulting from the anterior movement or removal of the IOZM is largely constant along the tooth row – around 10%. Therefore, it would seem that the extension of the zygomaticomandibularis on to the rostrum evolved in order to increase bite force (as proposed by Becht, 1953). Given that there is minimal difference in bite force between the models with the IOZM in its original position and those with the origin moved posteriorly, the position of the IOZM origin halfway along the rostrum may simply have coevolved with the lengthening of the rostrum in this species. Alternatively, Herrel et al. (2012) have suggested that the anterior insertion of the IOZM in L. aenigmamus increases the horizontal component of biting, which leads to anterior displacement of the mandible during jaw closing, which may be advantageous for the processing of leaves. Whatever the driving force behind the anterior extension of the IOZM, the results here indicate that the reason that the IOZM does not extend any further along the rostrum, as is seen in some other hystricomorph rodents such as the springhare (Offermans & De Vree, 1989) and capybara (Müller, 1933), is because this would lead to a reduction in bite force at both the incisors and cheek teeth (Table 4).

The results of this study have provided some important insights into the role of the IOZM muscle in the feeding behaviour of L. aenigmamus. Further investigations into other rodents, particularly other hystricomorphs, will enable us to understand whether the ability of the IOZM to increase bite force is unique to L. aenigmamus or common to all rodents that possess it. If the latter scenario is true, this could provide a selective advantage that may have driven the evolution of the IOZM, and could explain why it has evolved independently in several rodent groups (Ctenohystrica, Dipodidae, Anomaluroidea and Gliridae). However, finite element models can only shed light on static loading, and therefore cannot inform about the dynamic processes to which the IOZM may contribute, such as propalineal (antero-posterior) movements of the lower jaw, jaw opening or the fine control of gnawing. These activities can be addressed with dynamic modelling techniques, such as multibody dynamics analysis (e.g., Jones et al., 2012), and may well provide a fruitful avenue of research in the future.

We thank Jean-Pierre Hugot for providing access to the specimens and Dominique Adriaens, Loes Brabant and Luc Van Hoorebeke for scanning the specimens at the University of Ghent CT facility (UGCT). We are grateful to Paul O’Higgins and Michael Fagan for access to VOX-FE finite element software. Thanks are due to Laura Fitton for help with model construction, Peter Bazira for technical support with high-performance computing, and Andrew McIntosh and Thomas Püschel for assistance with GMM software.

Additional Information and Declarations

Competing Interests

Author Contributions

The authors declare that they have no competing interests.

Philip G. Cox conceived and designed the experiments, analyzed the data, wrote the paper.

Joanna Kirkham and Anthony Herrel performed the experiments, analyzed the data, wrote the paper.

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
