# Peer review of "Masticatory biomechanics of the Laotian rock rat, Laonastes aenigmamus, and the function of the zygomaticomandibularis muscle"

_PeerJ, doi:10.7717/peerj.160_

## Round 0.1 · original submission · Minor Revisions

Both reviewers believe this is an interesting and well-written article that combines 3DGM with FEM. Both reviewers request that the authors provide an illustration with attachment sites for the muscle(s) included in the models, and I encourage the authors to do so.
One reviewer has no other substantive comments, but the second challenges some of the specific conclusions -- and these should be addressed in the revision. I tend to agree that the unloaded model unduly impacts the ordination space, and could be excluded without loss of information. I also agree with the reviewer's comment about Figure 4. Please detail pecisely how you have responded to the various suggestions and queries in the revision response files (including rebuttals).

Reviewer 1 ·

Basic reporting

No comments

Experimental design

No comments

Validity of the findings

No comments

Additional comments

This is an excellent, well written article on a fascinating newly discovered species of rodent. I think that it would be very useful to include an additional figure that shows attachment sites and lines of action (or vectors) of the jaw adductor muscles used in the analyses.

One minor editorial note - line 137: Following Cox et al.........

Reviewer 2 ·

Basic reporting

No comments

Experimental design

No comments

Validity of the findings

No comments

Additional comments

This is a well written and interesting paper that uses a 3D analytical approach, combining both geometric morphometrics and finite element analysis, to understand the function of the Zygomaticomandibularis muscle.

I have some suggestions below that the authors should consider (particularly interpretation of the results in the discussion section), and I especially think the ms would benefit from an additional figure that shows precisely the origins and insertions of the muscles used to make the FEM – this would make the results entirely more easy to interpret for the reader.

Ln17: “specific diversity” should be “taxonomic diversity”
Ln78: insert, “e.g.” before “Dumont” ?
Ln104: “Cox et al.” typo, and same for the next ref.
Ln153: A ratio is expressed as two numbers separated by a colon e.g. “2:3”. It’s not the same as a fraction, percentage or proportion. The values expressed in Figure 3 relating to mechanical efficiency are incorrectly described as a ratio.
Ln185: Table 4 shows not demonstrates
Ln204: it is not clear why you included the unloaded model here – this clearly causes all of the variance in all other models to be clumped together on the right hand side of the plot, as a consequence of those all being very different from the unloaded model. Would it not be more informative to produce a plot excluding the unloaded model?
Ln209: maybe you should include a reference to Fig. 2 here (e.g. also shown in Fig. 2) after “region”
Ln209: insert “by PC2” at the end of the sentence
Ln214: looking at Fig. 2 what appears to actually be happening is dorsal movement (causing stress) of the area of the skull above the molars, i.e. between the orbits at the beginning of the calvaria surface. With procrustes superimposition, you spread the deformation across all your landmarks.
Ln221: “”indicating that even more deformation is occurring in these models” – so we conclude that the position of the IOZM, or at least the reason it’s there at all, is to limit the amount of stress in the skull (by limiting the amount of strain/displacement etc)
Ln222: this is not inferred
Ln226: insert “restrained” after loaded
Ln236: or the geometry of the zygomatic is adapted to restrain this?
Ln239: I wonder if this is likely? A system is adapted whereby expenditure from the temporalis would be required every time the masseter is used, otherwise the zygomatic gets snapped.
Ln244: should read “but is”
Ln248: this conclusion is difficult to evaluate without a figure showing us the FEM – i.e. the muscle insertions and origins
Ln254: I think this is an incorrect conclusion: look at Fig. 4, the position of attachment (again we need a figure with this) has little effect. BUT the presence or absence of IOZM is having an effect. Look at the orange symbols (no IOZM) in Fig 4, these are separate from the others because the absence of the IOZM is causing more PC shape change (which we know is a resultant of more strain = more stress). Mean VM stress might not pick this up because it's so vague and the pattern maybe similar (this depends again on where the other muscle attachments are, which we don't know) but this is likely controlled by skull morphology (mostly).
Ln259: But the reason it’s there, as shown in Fig. 4 is that it reduces landmark displacement/strain (and hence stress magnitude)
Ln263: but again, I think the data show that despite the IOZM being relatively small and having long muscle fibres, it is very effective at reducing the strain in the skull.
Ln264: this is because stress distribution is largely controlled by skull morphology in this case
Ln267: “reduces the bit force generated at all teeth” - but increases strain (Landmark displacement in the PCA, orange dots) i.e. removing the IOZM leads to a weaker bite AND a more strained/stressed skull. The mean stress and stress distribution may not appear different in Fig. 2, but the magnitudes of the stresses within this similar distribution must be higher with IOZM removed. This appears correct in Fig. 2 along the zygomatic arch, although images are inconclusive...
Ln268: again need a figure with muscle insertions…
Ln277: not sure that is correct - your data show that the presence of the IOZM evolved to increase bite force and decrease strain. Moving the IOZM has no result in Fig 4 on strain.

Figure 4: the small images of the skulls showing somehow deformation on the PC axes, are difficult to read – it’s hard to quickly and clearly see which regions of the skull are deforming – could the authors not add vectors to these graphics to show the direction of change, or a wireframe?

---

## Round 0.2 · accepted · Accept

The authors have responded quickly and thoroughly to reviewer suggestions, and I believe the ms. has been improved as a result of these additions and emendations. This study provides a fascinating window into the function of an evolutionary novelty, and I believe it merits publication in its revised form.